# Estimation of sodium consumption by novel formulas derived from random spot and 12-hour urine collection

**Pitchaporn Sonuch**[1], **Surasak Kantachuvesiri**[2,3]*, **Prin Vathesatogkit**[4], **Raweewan Lappichetpaiboon**[3], **Worawan Chailimpamontri**[3], **Nintita Sripaiboonkij Thokanit**[5], **Wichai Aekplakorn**[6]

**1** Department of Medicine, Faculty of Medicine Ramathibodi Hospital, Mahidol University, Bangkok, Thailand, **2** Division of Nephrology, Department of Medicine, Faculty of Medicine Ramathibodi Hospital, Mahidol University, Bangkok, Thailand, **3** Thai Lowsalt Network, The Nephrology Society of Thailand, Bangkok, Thailand, **4** Division of Cardiology, Department of Medicine, Faculty of Medicine Ramathibodi Hospital, Mahidol University, Bangkok, Thailand, **5** Ramathibodi Comprehensive Cancer Center, Faculty of Medicine Ramathibodi Hospital, Mahidol University, Bangkok, Thailand, **6** Department of Community Medicine, Faculty of Medicine Ramathibodi Hospital, Mahidol University, Bangkok, Thailand

* surasak.kan@mahidol.ac.th

**Data Availability Statement:** All relevant data are within the paper and its Supporting information files.

## Abstract

The gold standard for estimating sodium intake is 24h urine sodium excretion. Several equations have been used to estimate 24h urine sodium excretion, however, a validated formula for calculating 24h urine sodium excretion from 12h urine collection has not yet been established. This study aims to develop novel equations for estimating 24h urine sodium excretion from 12h and random spot urine collection and also to validate existing spot urine equations in the Thai population. A cross-sectional survey was carried out among 209 adult hospital personnel. Participants were asked to perform a 12h daytime, 12h nighttime, and a random spot urine collection over a period of 24 hours. The mean 24h urine sodium excretion was 4,055±1,712 mg/day. Estimated urine sodium excretion from 3 different equations using random spot urine collection showed moderate correlation and agreement with actual 24h urine sodium excretion (r = 0.54, P<0.001, ICC = 0.53 for Kawasaki; r = 0.57, P<0.001, ICC = 0.44 for Tanaka; r = 0.60, P<0.001, ICC = 0.45 for INTERSALT). Novel equations for predicting 24h urine sodium excretion were then developed using variables derived from 12h daytime urine collection, 12h nighttime urine collection, random spot urine collection, 12h daytime with random spot urine collection, and 12h nighttime with random spot urine collection which showed strong correlation and agreement with actual measured values (r = 0.88, P<0.001, ICC = 0.87; r = 0.83, P<0.001, ICC = 0.81; r = 0.67, P<0.001, ICC = 0.62; r = 0.90, P<0.001, ICC = 0.90; and r = 0.83, p<0.001, ICC = 0.82 respectively). Bland-Altman plots indicated good agreement between predicted values and actual 24h urine sodium excretion using the new equations. Newly derived equations from 12h daytime and 12h nighttime urine collection with or without casual spot urine collection were able to accurately predict 24h urine sodium excretion.

**Funding:** This study was financially supported by the Thai Health Promotion Foundation. The funder had no role in study design, data collection and analysis, decision to publish, or preparation of the manuscript.

**Competing interests:** The authors have declared that no competing interests exist.

## Introduction

Noncommunicable diseases (NCDs), such as cardiovascular disease (CVD), cancer, diabetes, and chronic respiratory diseases are the leading causes of death worldwide [1]. In Thailand, the top two causes of mortality are ischemic heart disease (IHD) and stroke. Over the past decade, the mortality rate has increased by 50% for IHD (from 21 to 32 per 100,000 population) and has more than doubled for stroke (from 21 to 48 per 100,000 population) [2]. Sodium is an essential nutrient for humans, but excessive sodium consumption is causally associated with high blood pressure [3]. Dietary sodium consumption of greater than the recommended daily amount of 5 grams of salt or 2,000 mg of sodium is a major risk factor for CVD-related mortality [4].

In 2010, the global mean sodium intake was 3.95 g/day (95% CI 3.89 to 4.01). This was nearly twice the WHO recommended limit of 2 g/day and equivalent to 10.06 (9.88–10.21) g/day of salt. Intakes were highest in East Asia, Central Asia and Eastern Europe (mean>4.2 g/day) and in Central Europe and Middle East/North Africa (3.9–4.2 g/day) [5]. In the UK and other developed nations, hypertension and its vascular complications were more common in ethnically African and South-Asian communities compared with Europeans [6, 7]. One important racial difference between ethnic groups is salt sensitivity and significantly suppressed activity of the renin–angiotensin–aldosterone system in African-origin hypertensive patients. As a consequence of this, they are more sensitive to a low-salt diet. There is also evidence that renin suppression is common in Japanese and Chinese hypertensive patients [8]. As demonstrated in a systematic review, sodium reduction from a high sodium intake level (201 mmol/day) to a level of 66 mmol/day resulted in a decrease in SBP/DBP of 1/0 mmHg in white participants with normotension and a decrease in SBP/DBP of 5.5/2.9 mmHg in white participants with hypertension. A few studies showed that the effects in black and Asian populations were greater [9].

Robust data on sodium consumption in the Thai population is limited [10]. Available studies are limited by method of measurement, selection bias and poor sampling. An estimate from a 24-hour dietary recall from the Thai national health examination survey IV, done in 2009, reported a daily sodium intake of 3264.5 mg/day [11]. However, this information may be subject to information bias due to over or under-reporting. A small study done in 200 patients, used 24-hour urine collection, the gold standard for estimating sodium intake, and reported a sodium intake of 2,955 mg per day. Other studies were done in small, non-representative samples such as hypertensive patients or patients with kidney disease [12, 13]. A recent study, the first nationally representative population-based survey using 24-hour urinary analyses indicated that dietary sodium consumption among Thai adults was 3,636 mg/day [14]. While national efforts in sodium reduction are underway, regular and accurate measurement of sodium intake is instrumental in monitoring progression towards goals of NCDs prevention and control [1].

There are several methods for assessment of sodium consumption including food frequency questionnaires, dietary recall, and 24-hour urine collection [15]. In general, 24-hour urine collection is regarded as the gold standard and is the most validated method for assessing sodium intake, as 90% of ingested sodium is excreted in the urine [16]. However, 24-hour urine collection is burdensome for participants and often results in incomplete data. Alternative methods have been derived to overcome this drawback, such as estimation of 24-hour urine sodium from spot urine collection [17]. Another study suggested that 12-hour urine collection at night may be more feasible and more reliable, as creatinine clearance correlated well with 24-hour urine samples [18], and 12-hour urine collection at night can be used to estimate 24-hour urine sodium excretion [19].

The purpose of our study was to validate previously published spot urine equations for predicting 24-hour urine sodium excretion using the Kawasaki, Tanaka, and INTERSALT equations (S1 Table) [20–22], and to establish novel formulas incorporating 12-hour urine collection and casual spot urine collection to predict 24-hour urine sodium excretion in Thai adults. More feasible methods and validated equations will be extremely beneficial for surveillance of sodium intake at the population level.

## Methods

### Study design and population

A cross-sectional survey was carried out among Ramathibodi Hospital personnel. All adults (male and female) over 18 years of age provided informed consent to participate in the study. Calculation of sample size was based on N4studies application (S1 Fig) [23]. The total number of subjects required for estimation of mean urine sodium of 3,600 and SD of 1,722 mg [14] for hospital personnel was 195. The study was conducted during July 2019 –September 2019. Participants who had a history of end stage renal disease, heart failure, cirrhosis, those who were started on a diuretic within 2 weeks, pregnant or breastfeeding women, those on contraceptive pills, women during menstruation and those unable to provide informed consent were excluded from the study. Details of the study were provided and informed consent was obtained from the participants. Participants collected 12-hour urine during the daytime, 12-hour urine during the nighttime, and a random spot urine collection. In addition, blood pressure was monitored, treatment and control of hypertension was assessed. The protocol was approved for ethics clearance by the Ethics Committee of Mahidol University.

### Data collection

Collection of demographics was carried out using a questionnaire. Data collected included age, sex, occupation, medical history and behavior related to dietary salt intake. Physical examination included measurement of height, weight, systolic and diastolic blood pressure. Physical measurements were taken immediately after the completion of demographic and behavioral information questionnaires. Blood pressure was taken by automatic blood pressure monitoring machine, Omron HEM-7130-L, after 15 minutes of rest, once participants had completed the questionnaire. Three blood pressure readings were recorded and calculated as mean systolic and diastolic blood pressure.

Each participant received two 3-liter containers for the 12-hour urine collections, one small container for spot urine collection, measuring cups and instructions on how to collect urine. After waking up, the first morning void was discarded. The participants were asked to collect all urine starting from the second void over a period of 12-hour during daytime. Then a 12-hour night time was collected until next morning including first morning urine. Participants were required to take note of the timing and the volume of urine collected with each void in the logbook provided. Participants were asked to randomly collect a spot urine during 12-hour collection using the small container. The urine samples were processed by the central laboratory of Ramathibodi Hospital for determination of urine volume, sodium, potassium and creatinine excretion.

Complete urine collection was defined as: total urine volume higher than 500 ml from 24-hour collection or not less than 250 ml from 12-hour collection and creatinine excretion more than 0.98 g/day for males and more than 0.72 g/day for females. Creatinine excretion was calculated using Roche/Hitachi cobas c laboratory system.

## Statistical analysis

Data analysis was carried out using STATA software, version 16.0. The results were expressed as mean ± standard deviation. Pearson correlation coefficient and Intraclass correlation coefficient (ICC) were calculated to assess the correlation between actual 24-hour urine sodium excretion and estimated 24-hour sodium using the Kawasaki, Tanaka and INTERSALT formulas (S1 Table). Student's t-test was used to compare two means and ANOVA test was used for comparison of three means or more. Multivariate linear regression analysis was used to create novel prediction equations for estimated 24-hour urine sodium excretion. The models incorporated multiple variables including age, height, weight, body mass index (BMI), urine sodium excretion, urine potassium excretion and urine creatinine excretion. Five novel equations estimating 24-hour urine sodium excretion were derived from: 1) 12-hour daytime urine collection, 2) 12-hour nighttime urine collection, 3) random spot urine collection, 4) 12-hour daytime collection with random spot urine collection, and 5) 12-hour nighttime collection with random spot urine collection. The models were analyzed separately for males and females. Pearson correlation coefficient and Intraclass correlation coefficient (ICC) were used to analyze the correlation between actual 24-hour urine sodium excretion and the estimated values calculated from the novel equations. The Bland-Altman method was used to estimate bias and agreement between equations. Statistical power was calculated based on an expected minimum $R^2$ value of 0.6, a sample size of at least 85, and a 0.05 alpha error. With 8 predictors, the power of the test was equal to 1.0.

## Results

Two hundred and fifty-two participants were recruited (Fig 1). After excluding those with incomplete urine collection based on volume and urine creatinine, those with incomplete data collection, and those who were pregnant, two hundred and nine individuals were included in the analysis. The mean age was 34.1 ± 10.8 years. There were more women (59.3%) than men (40.7%). Approximately 19.1% were hypertensives which was defined as systolic blood

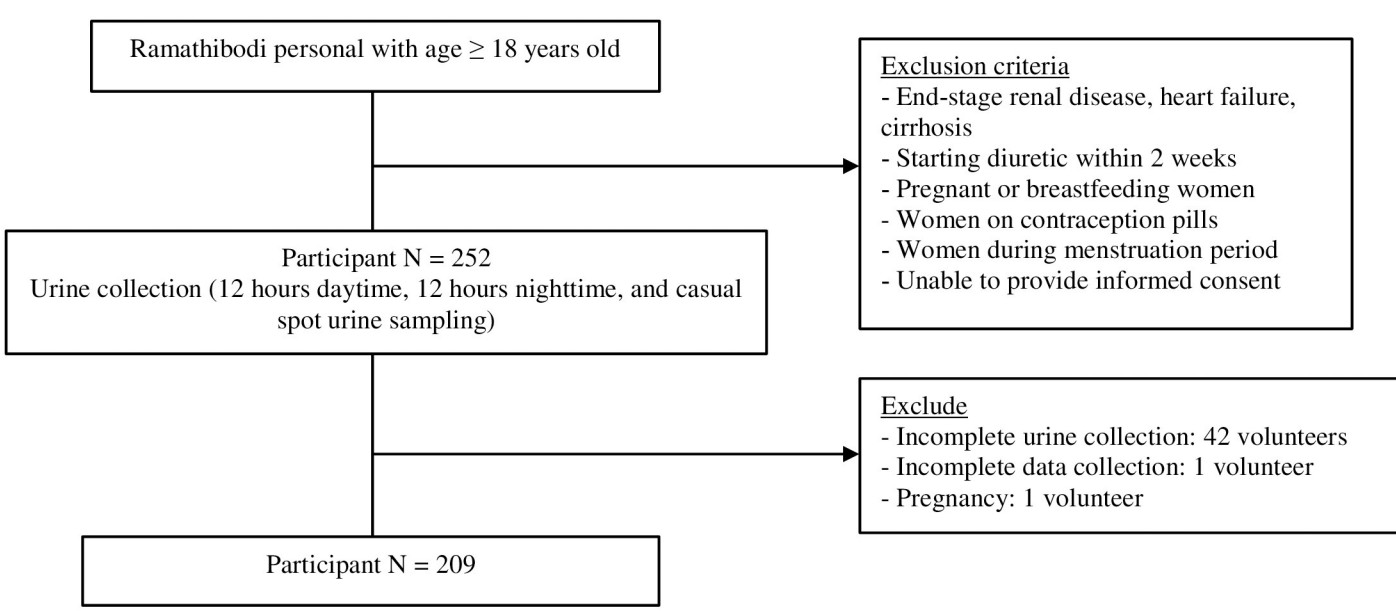

**Fig 1. Flow diagram of participants included in the study.**

**Table 1. Demographic data and clinical characteristic of the study population (N = 209).**

| Selected Characteristics | Mean ± SD, (%) |
| --- | --- |
| Age (years) | 34.1 ± 10.8 |
| Sex | |
| • Male | 85 (40.7%) |
| • Female | 124 (59.3%) |
| Occupation | |
| • Healthcare worker (HCW) | 98 (46.9%) |
| • Non-healthcare (non-HCW) | 111 (53.1%) |
| Weight (kg) | 66.1 ± 13.6 |
| Height (cm) | 164.5 ± 8.7 |
| BMI (kg/m$^2$) | 24.4±4.8 |
| BP (mmHg) | |
| • Systolic | 118 ± 15 |
| • Diastolic | 76 ± 10 |
| Hypertension (previous diagnosis or current use of hypertensive medications or SBP≥140mmHg or DBP≥90mmHg) | 40 (19.1%) |
| Other underlying diseases | |
| • Diabetic mellitus | 8 (3.8%) |
| • Dyslipidemia | 26 (12.4%) |
| • Chronic kidney disease | 3 (1.4%) |
| • Cardiovascular disease | 4 (1.9%) |
| • Others | 24 (11.5%) |
| 24-hour urine | |
| • Volume (ml) | 2,139 ± 908 |
| • Sodium excretion (mg/24h) | 4,055 ± 1,712 |
| • Potassium excretion (mg/24h) | 1,666 ± 619 |
| • Creatinine excretion (mg/24h) | 1,365 ± 439 |
| • Na/K ratio (mmol/mmol) | 4.4 ± 1.9 |
| Eating habits | |
| • Home cooking | |
| • 3 meals/day | 5 (2.4%) |
| • 0–2 meals/day | 204 (97.6%) |
| • Eating out | |
| • Convenience store | 156 (74.6%) |
| • Restaurant | 175 (83.7%) |
| • Shop/Street food | 166 (79.4%) |
| • Others | 25 (12.0%) |

pressure ≥ 140 mmHg or diastolic blood pressure ≥ 90 mmHg or current use of antihypertensive drugs with previous diagnosis of hypertension. In participants with hypertension, the number of patients receiving antihypertensive drugs were as follows: RAAS blockage 13 (Azilsartan 1, Enalapril 8, Losartan 4), calcium channel blocker 8 (Amlodipine 4, Manidipine 3, Lercanidipine 1), beta-blocker 2 (carvedilol 1, propranolol 1), hydralazine 1 and HCTZ 1. About 97.6% reported eating out at least one meal per day at different places such as restaurants (83.7%), shop/street food (79.4%), convenience stores (74.6%), and others (12.0%) (Table 1).

The mean sodium, potassium, and creatinine excretion from the 24-hour urine samples were 4,055 ± 1,712, 1,666 ± 619, and 1,365 ± 439 mg/day respectively. In subgroup analyses,

**Table 2. Urine excretion in subgroup population (N = 209).**

| Selected Characteristics (N) | Mean ± SD, (%) | | | | |
|---|---|---|---|---|---|
| | Volume (ml) | Urine sodium excretion (mg/24h) | Urine potassium excretion (mg/24h) | Urine creatinine excretion (mg/24h) | Urine Na/K ratio (mmol/mmol) |
| Age | | | | | |
| • 18–29 years (90) | 1,983 ± 831 | 3,517 ± 1,310 | 1,590 ± 573 | 1,355 ± 383 | 4.1 ± 1.8 |
| • 30–44 years (80) | 2,271 ± 1,015 | 4,593 ± 2,017 | 1,711 ± 676 | 1,414 ± 497 | 4.8 ± 2.1 |
| • 45–59 years (37) | 2,260 ± 817 | 4,285 ± 1,486 | 1,773 ± 599 | 1,308 ± 432 | 4.3 ± 1.6 |
| • ≥60 years (2) | 1,668 ± 131 | 2,487 ± 102 | 1,295 ± 21 | 927 ± 256 | 3.3 ± 0.1 |
| | P = 0.135 | P<0.001* | P = 0.308 | P = 0.304 | P = 0.055 |
| Sex | | | | | |
| • Male (85) | 2,108 ± 916 | 4,307 ± 1,694 | 1,679 ± 646 | 1,714 ± 410 | 4.8 ± 2.2 |
| • Female (124) | 2,161 ± 905 | 3,882 ± 1,710 | 1,657 ± 602 | 1,126 ± 263 | 4.3 ± 1.7 |
| | P = 0.684 | P = 0.078 | P = 0.806 | P<0.001* | P = 0.030* |
| Blood pressure | | | | | |
| • Hypertension (40) | 2,330 ± 793 | 4,592 ± 1,998 | 1,735 ± 625 | 1,406 ± 473 | 4.7 ± 1.8 |
| • Non-hypertension (169) | 2,094 ± 929 | 3,928 ± 1,618 | 1,650 ± 618 | 1,356 ± 431 | 4.3 ± 1.9 |
| | P = 0.140 | P = 0.027* | P = 0.434 | P = 0.516 | P = 0.238 |
| Occupation | | | | | |
| • HCW (98) | 2,031 ± 981 | 3,617 ± 1,406 | 1,646 ± 594 | 1,300 ± 374 | 4.0 ± 1.8 |
| • Non-HCW (111) | 2,235 ± 831 | 4,442 ± 1,865 | 1,683 ± 642 | 1,423 ± 484 | 4.7 ± 2.0 |
| | P = 0.106 | P<0.001* | P = 0.666 | P = 0.043* | P = 0.007* |

urine sodium excretion in males was higher than in females, 4,307 ± 1,694 versus 3,882 ± 1,710 mg/day respectively (P = 0.078). Non-health care workers (non-HCWs) had significantly higher urine sodium excretion than HCWs, 4,442 ± 1,865 versus 3,617 ± 1,406 mg/day respectively (P<0.001). Participants aged 30–44 years had the highest urine sodium excretion and significantly higher urine sodium excretion when compared to those aged 18–29 years, P<0.001. Moreover, those with hypertension had significantly higher urine sodium excretion than those without hypertension, 4,592 ± 1,998 versus 3,928 ± 1,618 mg/day respectively (P = 0.027) (Table 2).

Estimated 24-hour urine sodium excretion from spot urine collection using Kawasaki, Tanaka, and INTERSALT equations showed moderate correlation with actual 24-hour urine sodium excretion, r = 0.54, P<0.001, ICC = 0.53 (95%CI: 0.44 to 0.63); r = 0.57, P<0.001, ICC = 0.44 (95%CI: 0.34 to 0.55); and r = 0.60, P<0.001, ICC = 0.45 (95%CI: 0.35 to 0.56) respectively (Fig 2).

Multiple linear regression analyses were performed to identify factors that correlated with urine sodium excretion (S2 Table). Urine sodium excretion measured from 12-hour and random spot urine collection and spot urine creatinine significantly correlated with 24-hour urine sodium excretion.

Investigators then derived novel equations for estimating 24-hour urine sodium excretion (Table 3). Equations that were derived from 12-hour urine daytime and 12-hour urine nighttime collection showed strong correlation with 24-hour urine sodium excretion (r = 0.88, P<0.001, ICC = 0.87 (95%CI: 0.83 to 0.90); r = 0.83, P<0.001, ICC = 0.81 (95%CI: 0.76 to 0.85 respectively), whereas the equation derived from casual spot urine samples showed moderate correlation, r = 0.67, P<0.001, ICC = 0.62 (95%CI: 0.53 to 0.70) (Fig 2). Furthermore, when spot urine collection was added to the equation, these novel equations showed extremely strong correlation with 24-hour urine sodium excretion [r = 0.90, P<0.001, ICC = 0.90 (95%

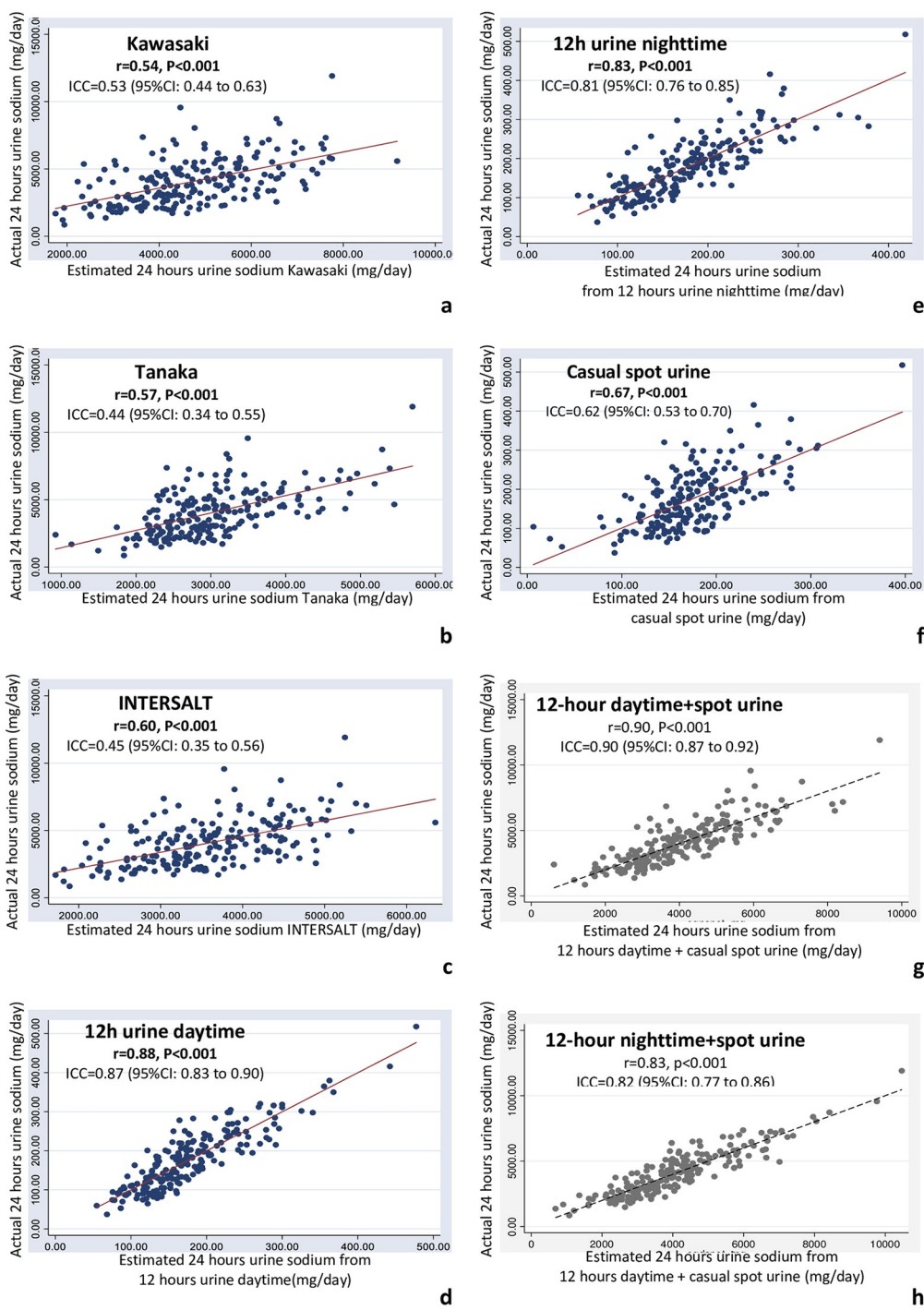

**Fig 2.** Correlation between the actual and the estimated equation 24 hours sodium excretion: (a) from Kawasaki equation, (b) from Tanaka equation, (c) from INTERSALT equation, (d) novel equation from 12-hour urine daytime collection, (e) novel equation from 12-hour urine nighttime collection, (f) novel equation from casual spot urine collection, (g) novel equation from 12-hour urine daytime collection with casual spot urine collection, (h) novel equation from 12-hour urine nighttime collection with casual spot urine collection.

**Table 3. Derived novel equation for estimating 24 hours urine sodium excretion.**

| Urine collection | Gender | Novel equation for estimated 24 hours urine sodium excretion (mg/day) |
|---|---|---|
| **12-hour daytime urine**<br>**R 0.88, p<0.001**<br>**ICC 0.87 (95%CI: 0.83 to 0.90)** | Male | 23*((Age*0.47)+(Wt*0.70)+(Ht*0.10)+(UNa daytime*1.07)-(UK daytime*0.10)+(UCr daytime*13.69)−9.54) |
| | Female | 23*((Age*0.69)+(Wt*0.96)-(Ht*0.01)+(UNa daytime*1.35)-(UK daytime*0.50)-(UCr daytime*35.64)−10.51) |
| **12-hour nighttime urine**<br>**R 0.83, p<0.001**<br>**ICC 0.81 (95%CI: 0.76 to 0.85)** | Male | 23*((Age*0.80)+(Wt*3.12)-(Ht*1.95)-(BMI*3.68)+(UNa nighttime*1.39)-(UK nighttime*1.07)-(UCr nighttime*34.05)+287.80) |
| | Female | 23*((Age*(-0.50))+(Wt*10.25)-(Ht*6.73)-(BMI*22.48)+(UNa nighttime*1.43)+(UK nighttime*0.26)-(UCr nighttime*50.48)+1092.97) |
| **Casual spot urine**<br>**R 0.67, p<0.001**<br>**ICC 0.62 (95%CI: 0.53 to 0.70)** | Male | 23*((Age*0.77)+(Wt*9.15)-(Ht*5.85)-(BMI*20.11)+(UNa spot*0.55)+(UK spot*0.19)-(UCr spot*0.41)+978.77) |
| | Female | 23*((Age*0.59)+(Wt*10.97)-(Ht*6.67)-(BMI*22.40)+(UNa spot*0.57)-(UK spot*0.08)-(UCr spot*0.44)+1068.44) |
| **12-hour daytime urine with casual spot urine**<br>**R 0.90, p<0.001**<br>**ICC 0.90 (95%CI: 0.87 to 0.92)** | Male | 23*((Age*0.15)+(BMI*1.74)+(UNa daytime*0.87)-(UK daytime*0.03)+(UCr daytime*39.17)+(UNa spot*0.20)+(UK spot*0.10)-(UCr spot*0.25)+25.71) |
| | Female | 23*((Age*0.67)+(BMI*1.32)+(UNa daytime*1.20)-(UK daytime*0.18)-(UCr daytime*7.51)+(UNa spot*0.34)-(UK spot*0.39)-(UCr spot*0.12)-2.14) |
| **12-hour nighttime urine with casual spot urine**<br>**R 0.83, p<0.001**<br>**ICC = 0.82 (95%CI: 0.77 to 0.86)** | Male | 23*((Age*0.31)+(BMI*5.26)+ (UNa nighttime*1.09)-(UK nighttime*1.01)-(UCr nighttime*0.80)+(UNa spot*0.26)+(UK spot*0.45)-(UCr spot*0.24)-43.74) |
| | Female | 23*((Age*(-0.63))+(BMI*2.76)+ (UNa nighttime*1.39)+(UK nighttime*0.05)-(UCr nighttime*39.65)+(UNa spot*0.08)+(UK spot*0.35)-(UCr spot*0.19)+41.92) |

Note: Age (years); Wt, weight (kg), Ht, height (cm), BMI, body mass index (kg/m²); UNa, urine sodium (mmol/12hr) from daytime and nighttime, UNa spot, spot urine sodium (mmol/L), UK, urine potassium (mmol/12hr) from daytime and nighttime, UK spot, spot urine potassium (mmol/L), UCr, urine creatinine (g/12hr) from daytime and nighttime, UCr spot, spot urine creatinine (mg/dL)

CI: 0.87 to 0.92) for 12-hour urine daytime collection with spot urine collection equation, and r = 0.83, p<0.001, ICC = 0.82 (95%CI: 0.7 7 to 0.86) for 12-hour urine nighttime with spot urine collection equation]. The novel equations that were derived from 12-hour urine collections demonstrated stronger correlation compared to the equations that were derived from spot urine collection.

A Bland-Altman analysis was performed to determine the difference between actual 24-hour urine sodium excretion and estimated urine sodium from novel equations. Results indicated good agreement between estimates from novel equations and actual 24-hour urine sodium excretion, with biases of 6.45 mg/day for 12-hour urine daytime collection, -19.53 mg/day for 12-hour nighttime collection, -20.60 mg/day for casual spot urine collection, -6.06 mg/day for 12-hour urine daytime with spot urine collection, and -19.96 mg/day for 12-hour urine nighttime with spot urine collection. For random spot urine equation, Bland-Altman analysis showed biases of 655.05 mg/day for Kawasaki equation, -478.85 mg/day for Tanaka equation, and -1023.37 mg/day for INTERSALT equation (Fig 3).

## Discussion

Previous studies have demonstrated correlation between 12-hour and 24-hour urine sodium and creatinine excretion [18, 19]. In addition, collecting urine during 12-hour periods may be more practical for subjects, hence may predict sodium intake more accurately. The objective

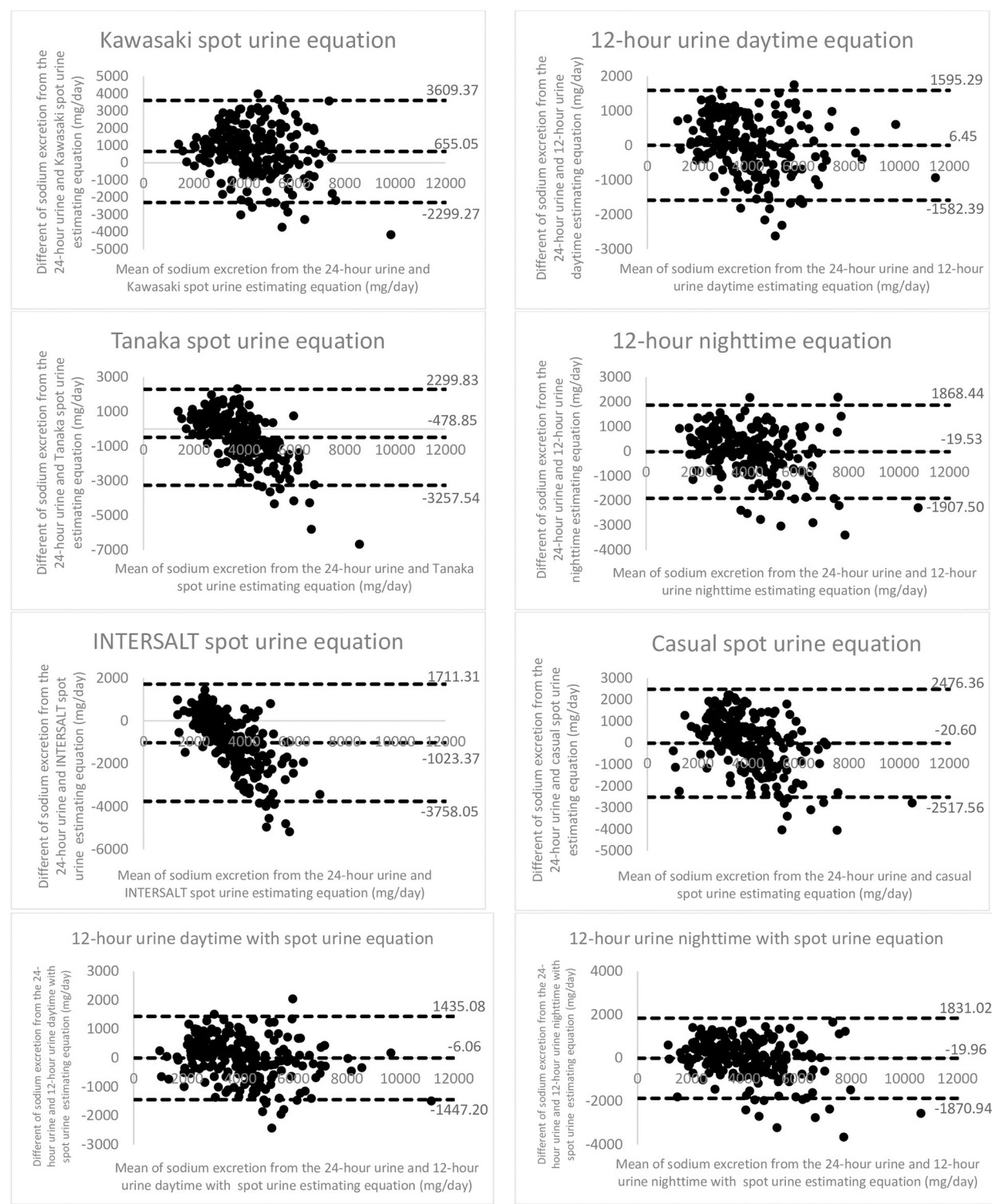

**Fig 3. Bland-Altman plot: The difference between 24-hour urine sodium excretion and estimated urine sodium from novel equations.** The dashed middle line represents the mean difference or bias. The other two dashed lines represents the 95% limits of agreement of the mean difference ± 1.96 standard deviation.

of our study was to investigate this correlation and formulate novel equations for estimating 24-hour urine sodium excretion from 12-hour urine collection, and casual spot urine collection. We were able to demonstrate that novel equations derived from 12-hour urine daytime and 12-hour urine nighttime collection with and without spot urine collection can be used to estimate 24-hour urine sodium excretion with good correlation. These novel equations demonstrate better correlation with actual urine sodium excretion than previously validated equations calculated from spot urine collection (Kawasaki, Tanaka and INTERSALT). A 12-hour urine collection may be more representative of a 24-hour urine collection than a spot urine collection, which is limited by diurnal variation in sodium excretion and is dependent on the amount and timing of sodium intake [24, 25].

In our study, we investigated the validity of previous equations that have been used to estimate urine sodium excretion in the Thai population from a spot urine collection. The Kawasaki and Tanaka equations were chosen for validation in this study because these formulas were derived from studies done in Asian participants, the Kawasaki equation using second morning void and the Tanaka equation using casual spot urine collection at different times during the day [20, 21] INTERSALT was also selected since it was derived from a large international population [22].

Results from our study suggest high sodium and low potassium intake in the enrolled subjects which is associated with an increased risk of NCDs and CVD. WHO recommends maximum daily intake of 5 grams of salt or 2,000 mg of sodium and at least 3,510 mg of potassium [4, 26, 27]. In our study, participants with hypertension had significantly higher urine sodium excretion than normotensives, affirming the association between dietary salt intake and hypertension, which is well-established in literature [28, 29]. Therefore, reduction in sodium intake may significantly decrease blood pressure in the hypertensive group. In contrast, urine potassium excretion was not significantly different in both groups. Furthermore, urine sodium excretion varied by age, with age groups of 30–44 and 45–59 having higher urine sodium excretion than the others. In line with our previous national surveys, sodium intake was higher among young people consuming higher calorie intake and fast food [14]. Our study demonstrated that non-HCWs had significantly higher urine sodium excretion than HCWs suggesting more knowledge and awareness among HCWs leading to their lower salt intake. Studies have reported that in the general population, knowledge of the health impacts of high salt intake is low [30]. Individuals with higher knowledge and awareness of the salt content and impact were significantly associated with lower salt intake [31, 32]. However, further research on the better understanding of salt knowledge and behavior in the population might facilitate the planning and implementation of a low salt intake program. Urine creatinine excretion in non-HCWs was also significantly higher than HCWs. This could be due to higher body mass index in the group of non-HCWs ($26.05\pm5.12$ kg/m$^2$ in non-HCW vs $22.64\pm3.45$ kg/m$^2$ in HCW, P<0.001) which affected urine creatinine excretion.

In participants with hypertension, some patients received antihypertensive drugs that may alter renal hemodynamics and sodium excretion. The administration of calcium-blocking drugs exerts a natriuretic response by exerting hemodynamic effects, as well as by acting directly on the proximal tubule and impairing sodium reabsorption in the distal tubule [33]. In contrast, beta-adrenergic antagonists have little or no clinical effect on glomerular filtration rate (GFR), urinary sodium or potassium excretion, free water clearance, or body fluid composition [34]. In animal models of salt sensitive hypertension, treatment with an angiotensin-converting-enzyme inhibitor (ACEI) or an angiotensin receptor blocker (ARB) effectively lowered blood pressure. In addition to lowering blood pressure, ACEI and ARB inhibition down-regulated ENaC and suppressed sodium reabsorption in renal tubules [35]. However, in our study, since the dosages of the antihypertensives were stable for at least 2 weeks before entering

the study, the effect of medications on spot urine sodium excretion and in turn, estimated urine sodium excretion will be trivial as patients are in a steady state. Furthermore, a prior study showed that medications such as diuretics and ACE inhibitor or angiotensin receptor blocker did not substantially affect the accuracy with which Na excretion was estimated by spot urine equation [36].

We validated various equations for predicting 24-hour urine sodium excretion from previously established equations and also formulated novel equations to estimated 24-hour urine sodium excretion in Thai participants. These new equations are derived from 12-hour urine collections to estimate 24-hour urine sodium excretion. They are robust and convenient for estimating sodium excretion and consumption in patients with uncontrolled hypertension. Furthermore, they provide a practical method to monitor sodium intake in clinical research or in epidemiological studies, especially in patients who work and can only collect urine for 12 hours at night time.

Strengths of this study include relatively complete and reliable urine samples. The main limitation is the study sample, which was hospital personnel, thereby limiting applicability to the general population. However, the study participants had similar characteristics to the general Thai population, therefore, the novel formulas for estimation of 24-hour urine sodium excretion should be applicable to the general Thai population with relatively good kidney function.

In summary, the newly derived equations from 12-hour urine daytime and 12-hour urine nighttime collection with or without casual spot urine collection may be applicable for estimation of 24-hour urine sodium excretion in the Thai population. This will be instrumental in the monitoring of sodium intake for NCDs prevention.

## Supporting information

**S1 Fig. N4studies application [23].**
(TIF)

**S1 Table. Three methods to estimated 24 hours urine sodium excretion from spot urine sample [20–22].**
(TIF)

**S2 Table. Multivariate linear regression analysis.**
(TIF)

**S1 Dataset.**
(XLSX)

## Acknowledgments

We would like to thank Ananthaya Kunjang for technical support and Dr. Pitchaphon Nissaisorakarn for editing the manuscript.

## Author Contributions

**Conceptualization:** Pitchaporn Sonuch, Surasak Kantachuvesiri, Worawan Chailimpamontri.

**Data curation:** Pitchaporn Sonuch, Surasak Kantachuvesiri, Raweewan Lappichetpaiboon.

**Formal analysis:** Pitchaporn Sonuch, Surasak Kantachuvesiri, Nintita Sripaiboonkij Thokanit.

**Investigation:** Pitchaporn Sonuch, Surasak Kantachuvesiri.

**Methodology:** Pitchaporn Sonuch, Surasak Kantachuvesiri, Prin Vathesatogkit.

**Project administration:** Pitchaporn Sonuch, Surasak Kantachuvesiri.

**Supervision:** Surasak Kantachuvesiri, Wichai Aekplakorn.

**Validation:** Pitchaporn Sonuch, Surasak Kantachuvesiri.

**Visualization:** Pitchaporn Sonuch, Surasak Kantachuvesiri.

**Writing – original draft:** Pitchaporn Sonuch.

**Writing – review & editing:** Pitchaporn Sonuch, Surasak Kantachuvesiri.

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
