## [Decision Letter · Decision Letter 0]

27 Apr 2021

PONE-D-21-09293

Estimation of sodium consumption by novel formulas derived from random spot and 12-hour urine collection

PLOS ONE

Dear Dr. Kantachuvesiri, 

Thank you for submitting your manuscript to PLOS ONE. After careful consideration, we feel that it has merit but does not fully meet PLOS ONE’s publication criteria as it currently stands. Therefore, we invite you to submit a revised version of the manuscript that addresses the points raised during the review process.

My observations to be answered:

- The study added new information to existing knowledge of Kawasaki, Tanaka, and Intersalt equations.

- In the introduction the authors made a literature revision, but there were no reported differences between handling of sodium in the Western, Eastern and Black populations.

-- In Methods is important to provide the calculation power performed for the inclusion of patients in the study.  The urine collection period was not described when the authors refer to the day or night period and also is important to know the start or end time of it. It is important describe the drugs that the volunteers normally received.

- The study analyzed other parameters in results as ages, gender, occupation, medical history, behavior towards dietary salt intake,  hypertension and other parameters that were described in results, however, such analyzes were a little out of the objective of the study. The drugs received by patients need to be discussed with regard to the influence on the handling of sodium.

- Figures need to be better worked, with better resolution.

- In the discussion, it would be important to emphasize the importance of the 12-hour urine collection findings in relation to the 24-hour urine in established calculations and its application in the future in the clinic.

- Native English proofreading is required

We look forward to receiving your revised manuscript.

Kind regards,

Dulce Elena Casarini, PhD, FAHA

Academic Editor

PLOS ONE

Additional Editor Comments:

The authors described in the manuscript equations/formulas to validated formula for predicting 24h urine sodium excretion from 12h urine collection. The study added new information to existing knowledge of Kawasaki, Tanaka, and Intersalt equations.

In the introduction the authors made a literature revision, but there were no reported differences between handling of sodium in the Western, Eastern and Black populations.

In Methods is important to provide the calculation power performed for the inclusion of patients in the study. The urine collection period was not described when the authors refer to the day or night period and also is important to know the start or end time of it. It is important describe the drugs that the volunteers normally received.

The study analyzed other parameters in results as ages, gender, occupation, medical history, behavior towards dietary salt intake, hypertension and other parameters that were described in results, however, such analyzes were a little out of the objective of the study. The drugs received by patients need to be discussed with regard to the influence on the handling of sodium.

Figures need to be better worked, with better resolution.

In the discussion, it would be important to emphasize the importance of the 12-hour urine collection findings in relation to the 24-hour urine in established calculations and its application in the future in the clinic.

Journal Requirements:

[This study was financially supported by the ThaiHealthPromotionFoundation.]

 [NO, The funders had no role in study design, data collection and analysis, decision to publish, or preparation of the manuscript.]

Reviewers' comments:

Reviewer's Responses to Questions

**Comments to the Author**

1. Is the manuscript technically sound, and do the data support the conclusions?

Reviewer #1: Yes

Reviewer #2: Partly

2. Has the statistical analysis been performed appropriately and rigorously? 

Reviewer #1: Yes

Reviewer #2: Yes

3. Have the authors made all data underlying the findings in their manuscript fully available?

Reviewer #1: Yes

Reviewer #2: Yes

4. Is the manuscript presented in an intelligible fashion and written in standard English?

Reviewer #1: No

Reviewer #2: Yes

5. Review Comments to the Author

Reviewer #1: To authors

The current manuscript is interesting and important. It involves prediction about cardiovascular diseases in Thailand through getting sodium urine measurement. The authors carried out urine tests and created new formulas with correlation between former formulas by other researchers and the new formula.

My questions:

Methods

I have not found quotes about participants with heart failure or cirrhosis. These syndromes cause low perfusion, secondary hyperaldosteronism. It can interfere with renal sodium excretion.

Approximately 19 % were hypertensives, but I have also found no reports of the use of drugs that block the renin-angiotensin system in hypertensive volunteers. Could these medicines interfere with kidney sodium excretion?

I am not sure or I didn't pay attention when reading the manuscript. What was the exact moment when each participant performed the spot urine collection? Was it before, during or after the 12h urine collection?

Results

What statistical test did you use to compare urinary sodium excretion between HCW and non-HCW groups? And about comparison between age? And hypertesion vs. non-hypertension? Have you performed with the Student's t test?

Discussion

In my opinion, you should report the primary result found in the in the first paragraph in discussion of the manuscript

Authors must perform a detailed review of English.

Reviewer #2: Soumach's article and collaborators describe new equations to predict 24-hour urinary sodium excretion. With the equations proposed by the authors, the collection of 12-hour urine, whether collected during the day or night, associated or not with spot urine, correlates significantly with studies that investigated sodium excretion. The study of Soumach and collaborators adds information to existing knowledge.

Some suggestions for improving the study

• I suggest that the specificity of the population studied is included in the title. Possibly, most (if not all) of the study participants were Eastern. It is known, for example, that black individuals have a distinct handling of sodium. Even in the introduction of the work the authors refer only to the sodium intake of the Thai population.

• Although the authors elegantly demonstrate that the proposed formulas predict and validate excretion in a shorter time (12 hours) it would not be important for the authors to evidence the gains in the use of these equations. The discussion of the benefit of using equations takes place in only one paragraph.

• How did the authors calculate the number of patients included in each group? Which test was used to calculate the sample?

• When the authors refer to the day or night period, the start or end time of the collection is not clear.

• Could the fact that patients answer the questionnaire while resting could not determine the increase in blood pressure?

• Antihypertensive drugs that alter renal hemodynamics (such as calcium channel blockers) or aldosterone secretion could not interfere with the results?

• The objectives of the work were related to the development of the equations. However, the work also makes comparations between ages, gender, the fact that patients are hypertensive or health professionals. The focus of the work seems to me scattered. It would be interesting, the correlation between 24-hour and 12-hour measurements in the different strata proposed by the authors.

• What was the statistical analysis software used by the authors?

• In equations, I think it should be sororded "UNa" instead of "Una". UK is capitalized

6. PLOS authors have the option to publish the peer review history of their article (what does this mean?). If published, this will include your full peer review and any attached files.

Reviewer #1: No

Reviewer #2: No

---

## [Author Response · Author response to Decision Letter 0]

24 May 2021

Response to Reviewer

- In the introduction the authors made a literature revision, but there were no reported differences between handling of sodium in the Western, Eastern and Black populations.

Ans: We have added the literature review in the introduction part as follows:

In 2010, the global mean sodium intake was 3.95 g/day (95% CI 3.89 to 4.01). This was nearly twice the WHO recommended limit of 2 g/day and equivalent to 10.06 (9.88–10.21) g/day of salt. Intakes were highest in East Asia, Central Asia and Eastern Europe (mean>4.2 g/day) and in Central Europe and Middle East/North Africa (3.9-4.2 g/day).(5) In the UK and other developed nations, hypertension and its vascular complications were more common in ethnically African and South-Asian communities compared with Europeans.(6,7) One important racial difference between ethnic groups is salt sensitivity and significantly suppressed activity of the renin–angiotensin–aldosterone system in African-origin hypertensive patients. As a consequence of this, they are more sensitive to a low-salt diet. There is also evidence that renin suppression is common in Japanese and Chinese hypertensive patients.(8) As demonstrated in a systematic review, sodium reduction from a high sodium intake level (201 mmol/day) to a level of 66 mmol/day resulted in a decrease in SBP/DBP of 1/0 mmHg in white participants with normotension and a decrease in SBP/DBP of 5.5/2.9 mmHg in white participants with hypertension. A few studies showed that the effects in black and Asian populations were greater.(9) (Paragraph2, Page3)

- In Methods is important to provide the calculation power performed for the inclusion of patients in the study. 

Ans:

We have added a sentence describing the calculation of power in the statistical analysis section as follows: Power of test was assessed based on an expected minimum value of R squared of 0.6, sample size of at least 85, alpha error 0.05 and 6 predictors, the power of test was equal to 1.0. (Statistical analysis, Page7)

- The urine collection period was not described when the authors refer to the day or night period and also is important to know the start or end time of it. 

Ans:

Regarding to urine collection period, the first morning void was discarded. A 12-hour urine daytime started from the second void and collected over a period of 12-hours, Then a 12-hour night time was collected until next morning including first morning urine. (Paragraph2, Page6)

- It is important describe the drugs that the volunteers normally received.

Ans: We have added the information on drug used and discussed in the discussion part as follows:

In subgroup of participants with hypertension, numbers of patients receiving antihypertensive drugs were followings RAAS blockage (Azilsartan 1, Enalapril 8, Losartan 4), calcium channel blocker (Amlodipine 4, Manidipine 3, Lercanidipine 1), beta-blocker (carvedilol 1, propranolol 1), hydralazine in 1 and HCTZ in 1. (Paragraph2, Page13)

- The study analyzed other parameters in results as ages, gender, occupation, medical history, behavior towards dietary salt intake, hypertension and other parameters that were described in results, however, such analyzes were a little out of the objective of the study. The drugs received by patients need to be discussed with regard to the influence on the handling of sodium

Ans:

In participants with hypertension, some patients received antihypertensive drugs that may alter renal hemodynamics and sodium excretion. The administration of calcium-blocking drugs exerts a natriuretic response by exerting hemodynamic effects, as well as by acting directly on the proximal tubule and impairing sodium reabsorption in the distal tubule.(30) In contrast, beta-adrenergic antagonists have little or no clinical effect on glomerular filtration rate (GFR), urinary sodium or potassium excretion, free water clearance, or body fluid composition.(31) In animal models of salt sensitive hypertension, treatment with an angiotensin-converting-enzyme inhibitor (ACEI) or an angiotensin receptor blocker (ARB) effectively lowered blood pressure. In addition to lowering blood pressure, ACEI and ARB inhibition downregulated ENaC and suppressed sodium reabsorption in renal tubules.(32) However, in our study, since the dosages of the antihypertensives were stable for at least 2 weeks before entering the study, the effect of medications on spot urine sodium excretion and in turn, estimated urine sodium excretion will be trivial as patients are in a steady state. Furthermore, a prior study showed that medications such as diuretics and ACE inhibitor or angiotensin receptor blocker did not substantially affect the accuracy with which Na excretion was estimated by spot urine equation.(33 (Paragraph2, Page13)

- Figures need to be better worked, with better resolution.

Ans:

We have revised the figures as suggested

- In the discussion, it would be important to emphasize the importance of the 12-hour urine collection findings in relation to the 24-hour urine in established calculations and its application in the future in the clinic.

Ans:

We added discussion on the importance of the new 12-hour urine equation and its application on page 14

- Native English proofreading is required

Ans:

Native English proofreading was performed

---

## [Decision Letter · Decision Letter 1]

24 Aug 2021

PONE-D-21-09293R1

Estimation of sodium consumption by novel formulas derived from random spot and 12-hour urine collection

PLOS ONE

Dear Dr. Kantachuvesiri

Thank you for submitting your manuscript to PLOS ONE. After careful consideration, we feel that it has merit but does not fully meet PLOS ONE’s publication criteria as it currently stands. Therefore, we invite you to submit a revised version of the manuscript that addresses the points raised during the review process.

The review was adequate and two small points would need to be reviewed, topics that are in line with those also highlighted by one of the referees:

-  the statistical analysis to Table 2 when the authors discuss age, 

-  in discussion explain the difference between HCW and non-HCW in excretions of sodium and creatinine. 

We look forward to receiving your revised manuscript.

Kind regards,

Dulce Elena Casarini, PhD, FAHA

Academic Editor

PLOS ONE

Journal Requirements:

Additional Editor Comments (if provided):

All changes made were appropriate in this new version, and I suggest that the authors observe the note made by one of the referees about adding the statistical analysis to Table 2 when the authors discuss age. Regarding the discussion, the difference between HCW and non-HCW in sodium and creatinine excretions was not highlighted, which was also observed by one of the referees.

With these minor corrections the article is in the profile for publication.

Reviewers' comments:

Reviewer's Responses to Questions

**Comments to the Author**

1. If the authors have adequately addressed your comments raised in a previous round of review and you feel that this manuscript is now acceptable for publication, you may indicate that here to bypass the “Comments to the Author” section, enter your conflict of interest statement in the “Confidential to Editor” section, and submit your "Accept" recommendation.

Reviewer #1: All comments have been addressed

Reviewer #2: All comments have been addressed

2. Is the manuscript technically sound, and do the data support the conclusions?

Reviewer #1: Yes

Reviewer #2: Partly

3. Has the statistical analysis been performed appropriately and rigorously? 

Reviewer #1: Yes

Reviewer #2: Yes

4. Have the authors made all data underlying the findings in their manuscript fully available?

Reviewer #1: Yes

Reviewer #2: Yes

5. Is the manuscript presented in an intelligible fashion and written in standard English?

Reviewer #1: Yes

Reviewer #2: Yes

6. Review Comments to the Author

Reviewer #1: Dear authors

The current manuscript is interesting and important. It involves prediction about cardiovascular diseases through getting urine sodium measurement.

All comments have been addressed.

Reviewer #2: The study by Sonouch and collaborators brings gains from its original version and is much better organized.

The article actually brings two studies, the first that discusses the sodium excretion of 209 Thai participants. The second is the design of equations that increase the correlation between urine collection for 12 hours (daytime or nighttime), associated or not with the isolated urine sample. The correlations when associated with the 12-hour sample when associated with the isolated sample present a high degree of correlation against the gold standard, which is the 24-hour sodium dosage.

Because the study is local, I strongly recommend that the study title contain this information. The stratified analysis of the information, verified in table 2, was a gain for the study, but the statistical analysis was lacking in the table when the authors discuss age. I also did not see in the article discussion why the difference between HCW and non-HCW in sodium and creatinine excretions.

I still believe that extrapolating mathematical behaviors in a small sample should be taken with caution. The authors do not demonstrate calculations the estimate of the necessary population.

7. PLOS authors have the option to publish the peer review history of their article (what does this mean?). If published, this will include your full peer review and any attached files.

Reviewer #1: No

Reviewer #2: No

---

## [Author Response · Author response to Decision Letter 1]

21 Oct 2021

Response to Reviewer

1. The statistical analysis to Table 2 when the authors discuss age

Ans:

We have added ANOVA test was used for comparison of three means or more. (Paragraph1, Page7) and inserted P value in Table2.

2. In discussion explain the difference between HCW and non-HCW in excretions of sodium and creatinine.

Ans:

 We have add discussion in Paragraph2, Page13

Furthermore, urine sodium excretion varied by age, with age groups of 30-44 and 45-59 having higher urine sodium excretion than the others. In line with our previous national surveys, sodium intake was higher among young people consuming higher calorie intake and fast food.(14) Our study demonstrated that non-HCWs had significantly higher urine sodium excretion than HCWs suggesting more knowledge and awareness among HCWs leading to their lower salt intake. Studies have reported that in the general population, knowledge of the health impacts of high salt intake is low.(30) Individuals with higher knowledge and awareness of the salt content and impact were significantly associated with lower salt intake.(31,32) However, further research on the better understanding of salt knowledge and behavior in the population might facilitate the planning and implementation of a low salt intake program. Urine creatinine excretion in non-HCWs was also significantly higher than HCWs. This could be due to higher body mass index in the group of non-HCWs (26.05±5.12 kg/m2 in non-HCW vs 22.64±3.45 kg/m2 in HCW, P<0.001) which affected urine creatinine excretion.

---

## [Editor Report · Decision Letter 2]

10 Nov 2021

Estimation of sodium consumption by novel formulas derived from random spot and 12-hour urine collection

PONE-D-21-09293R2

Dear Dr. Kantachuvesiri

We’re pleased to inform you that your manuscript has been judged scientifically suitable for publication and will be formally accepted for publication once it meets all outstanding technical requirements.

Kind regards,

Dulce Elena Casarini, PhD, FAHA

Academic Editor

PLOS ONE

Additional Editor Comments (optional):

The authors answered all the last questions asked by the referees that improved the manuscript. This latest revision has been carefully done, and therefore the manuscript is accepted for publication.

Reviewers' comments:

The authors answered all the last questions asked by the referees that improved the manuscript. This latest revision has been carefully done, and therefore the manuscript is accepted for publication.

---

## [Editor Report · Acceptance letter]

22 Nov 2021

PONE-D-21-09293R2 

Estimation of sodium consumption by novel formulas derived from random spot and 12-hour urine collection 

Dear Dr. Kantachuvesiri:

I'm pleased to inform you that your manuscript has been deemed suitable for publication in PLOS ONE. Congratulations! Your manuscript is now with our production department. 

Kind regards, 

on behalf of

Dr. Dulce Elena Casarini 

Academic Editor

PLOS ONE